# What Is the Relationship between Natural Protected Areas and Stakeholders? Based on Literature Analysis from 2000–2021

Yangyang Zhang [1], Jiaoyang Xu [1], Yunong Yao [2], Zhaogui Yan [1], Mingjun Teng [1] and Pengcheng Wang [1,*]

1 College of Horticulture and Forestry Sciences, Hubei Engineering Technology Research Center for Forestry Information, Huazhong Agricultural University, No. 1 Shizishan Street, Hongshan District, Wuhan 430070, China; yangyangzhang1993@outlook.com (Y.Z.); hzau1994xjy@163.com (J.X.); gyan@mail.hzau.edu.cn (Z.Y.); tengmingjun@hotmail.com (M.T.)
2 State Key Laboratory of Urban and Regional Ecology, Research Center for Eco-Environmental Sciences, Chinese Academy of Sciences, No. 18 Shuangqing Road, Haidian District, Beijing 100085, China; yaoyn331@163.com
* Correspondence: wangpc@mail.hzau.edu.cn

**Abstract:** The establishment of natural protected areas (NPAs) is an effective means to deal with the degradation of ecosystems caused by climate change and human activities. The area and number of NPAs in the world have shown an obvious growth trend, and their development has ushered in a new bottleneck. More importantly, the management quality of NPAs should be improved, and the key to improving management quality lies in human beings, but the stakeholder groups involved in NPAs are often overlooked by policymakers. In this study, a quantitative review of the global scientific literature on NPAs stakeholders was conducted using a bibliometric approach. The research hotspots and trends, number, time, and countries were analyzed based on data from published articles. The stakeholder types and internal relationships in NPAs were summarized and mapped. The common problems of resources and community resident management among stakeholders were discussed. A total of 5584 research articles selected from the Web of Science core collection database were used as data sources and were visualized using VOSviewer and the Biblioshiny program in the R language. The results of the study help to reveal the mutual influence mechanism between stakeholders during the development of nature reserves and contribute to the sustainable development of global protected areas and human well-being.

**Keywords:** natural protected areas; stakeholders; bibliometric analysis; community participation

## 1. Introduction

Natural protected areas (NPAs) are the first line of defense for biodiversity conservation globally. NPAs provide multiple benefits to humans and are critical to protecting species threatened by land-use change and habitat loss [1,2]. Since 2010, over 21 million km$^2$ has been placed under the protected and conserved areas, meaning that 42% of the area now within protected and conserved areas has been added in the last decade [3]. However, the development of NPAs continues to be challenged by unpredictable factors, particularly human factors, with approximately one third of NPAs being affected by intensive human activity [4]. The lessons of the past 10 years should be taken forward into the next decade, paving the way for more effective and equitable systems of protected and conserved areas [3]. The Convention on Biological Diversity (CBD, COP15) meeting to be held in Kunming, China in 2021, emphasized that humans and nature are a community of life, humans should respect nature, conform to nature and protect nature. At the meeting, it was proposed to achieve sustainable use and benefits of biodiversity by 2050 to share and realize the beautiful vision of "harmonious coexistence between man and nature". Therefore, clarifying the relationship and common problems between NPAs and stakeholders

is a key prerequisite for realizing the harmonious coexistence of humans and nature and protecting biodiversity.

NPAs are considered to be the most effective conservation policy tools to achieve a more sustainable green development [5], of which the main goals are to maintain biodiversity, ecological processes and ecosystem services, which are the nature and key strategies to halt biodiversity loss [6,7]. NPAs have multiple goals and functions, are closely related to human production and living space, and involve multiple stakeholders [8–12]. In the current context of climate change and the COVID-19 pandemic, the economy has collapsed, dramatically changing social lives, the driving factors affecting the functioning of NPAs are increasingly focused on human activities, especially in developing or impoverished areas [13].

Some of the conflicts between conservation and development in NPAs lie in the achievement of whether the natural resources can be reasonably protected and utilized; the core of which is the coordination of the relationship between people and natural resources, which face numerous threats mainly from human-driven factors [14–16]. When it comes to using resources in and around NPAs, there are many stakeholders involved, including local residents, resource users, and local government managers [11]. The complex ownership status of land and sea areas affects the rights and interests of all stakeholders in the process of governance goals of NPAs [17,18]. When interest demands are not dealt with in a timely manner, it will lead to further conflicts and cause the destruction of resources in the NPAs [19]. Simply relying on the establishment of NPAs to achieve the goals of ecological protection and biodiversity maintenance will not work. It is therefore necessary to resolve conflicts between stakeholders and nature. Thus, the NPAs should not only take into account ecological protection, but also promote the sustainable development of the regional economy and the improvement of the life for local residents [20].

Because most NPAs are located in remote areas [21], restrictions to the development of natural resources may further exacerbate the existing poverty [14], so there is a conflict between achieving conservation goals and reducing or eradicating poverty in these areas [22]. Due to the economic backwardness of poverty-stricken areas, the livelihood of the residents in the surrounding communities of the NPAs is often relying entirely on natural resources, resulting in frequent violent confrontations between the residents and the NPAs managers [23]. Activities such as deforestation, and illegal hunting and fishing [24–26] will directly or indirectly adversely affect NPAs, resulting in the decline in forest vegetation cover, reduction in wildlife habitat, reduction in biodiversity, and the loss of soil and water. In addition, what cannot be ignored is the relationship between the community residents living in and around the NPAs and wild animals. The conflict between the two will have a greater impact on the livelihood of the local community and may become an important factor in the continuous maintenance of NPAs [23,27]. Therefore, it is necessary to adopt a scientific approach to the management of NPAs, and to address as many issues as possible in the interest of all stakeholders to ensure better development of the local economy and improvement in the livelihood of the local communities.

It is expected that the scientific literature on NPAs conservation and economic development will continue to increase in the coming years. The development of NPAs should not only stop at the increase in area and quantity but also improve the management quality of NPAs. As the range of disciplines involved in scientometrics analysis has expanded in recent years [28], surveying the scientific literature is an important first step in understanding past research and achievements and predicting future scientific trajectories. In this regard, the combination of social network analysis and bibliometrics has proven to be a useful tool for quantitatively assessing patterns and trends in the scientific literature [29–32].

In this review, we aimed to analyze the existing research literature on NPAs and present a comprehensive bibliometric summary of the research status, research hotspots and regional distribution, and visualize them to determine the development trend of past and present research fields. Based on these analyses, we put forward the types and

composition characteristics of stakeholders and the main contradictions between NPAs and economic development.

## 2. Materials and Methods—Data Collection and Processing

### 2.1. Data Collection

To achieve the objective of the study, a bibliometric analysis of the global scientific literature on NPAs stakeholders was conducted. Although the CBD has given the international community a broad definition of what an NPA is, there is no definitive definition for NPAs, and there is no agreed international schema for all NPAs [33]. In order to ensure the comprehensiveness of the research object, the guidelines for applying NPAs management categories included "IUCN WCPA" best practice guidance on recognizing NPAs and assigning management categories and governance types (https://www.iucn.org/theme/protected-areas/about/protected-area-categories (accessed on 22 April 2022)), and describe in as much detail as possible the types of NPAs that may exist. Because these categories are recognized by international bodies such as the United Nations and by many national governments as the global standard for defining and classifying NPAs and as such are increasingly being incorporated into government legislation. Therefore, we used as many NPAs types as possible in the search topic, to better study the global NPAs types.

We collected from the Web of Science (WOS) core collection database, the topic (Search for titles, abstracts, author keywords, and Keywords Plus) of the text, the retrieval logical relationship was "intersection" (AND), the topic was "Stakeholder*" AND "Protected Area*/Natural Park*/Nature Reserve*/Wildlife Reserve*/National Park*/Natural Monument*/Habitat Management Area*/Species Management Area*/Protected Landscape or Seascape*/Scientific Reserve*/Natural Landmark*/Biosphere Reserve*/Bird Sanctuary*/Natural Biotic Reserve*/Fauna and Flora Reserve*". The time span was from 1 January 2000 to 31 December 2021, a total of 15 searches were carried out, and a total of 5584 unique articles were obtained.

We decided to extract these document types because they are considered peer-reviewed "certified knowledge". After selecting the "Title, Source Publication, Abstract, Year of Publication" option, the metadata of the WOS database research were exported as a "text" file and imported into VOSviewer (version 1.6.13) [34]. At the same time, we used "Full records and cited references", as the selection criteria and exported the file as BibTex text so that can be recognized by R software [28].

### 2.2. Data processing

VOSviewer (https://www.vosviewer.com/ (accessed on 22 April 2022)) is a free bibliometric analysis and visualization software [35] that is powerful in "co-occurrence" network clustering as well as density analysis, generating network graphs that assign the positions of displayed items accordingly based on their degree of association. Strongly correlated items are close to each other, while weakly correlated items are farther apart. To investigate trends in major themes relevant to the conservation stakeholder study, a co-occurrence map of themes was generated. Keywords co-occurrence networks quantitatively represent the connection between two keywords based on the number of publications in which two keywords appear simultaneously in the title, abstract, or keyword list. In order to visualize the most important keywords, in the choose fields in VOSviewer, select the frequency of title and abstract fields, check "Ignore structured abstract labels" and "Ignore copyright statements". To avoid duplication in the generated network diagram, and to display the results correctly, a synonym file (replace or merge) that unifies the keywords related to the same topic is used, and the new file is reimported for operation. In addition, VOSviewer's overlay visualization application allows network items to be represented on a temporal gradient, representing the co-occurrence of network items as a function of time, this visualization is based on the average publication year of documents in which keywords appear [32]. For better visualization, the number of occurrences of



words in the title and abstract was set to exceed 320 in VOSviewer, and there were a total of 89 words. For synonym replacement or merging operations, invalid words were ignored, and 60 high-frequency keywords were finally obtained.

We used the R software for bibliometric analysis (version 4.1.2) [36], mainly using the Bibliometrix R-package (http://www.bibliometrix.org (accessed on 25 April 2022)). It provides a set of tools for quantitative research in bibliometrics and written in the R language. Users which can perform relevant measurement visualization analysis in the web interface. It has a variety of functional modules, and can also be represented by intuitive visualization or maps (such as two-dimensional maps, dendrograms, and social networks). It is also free and open-source, and can be used on the web [28]. In order to investigate the publication volume and mutual cooperation status of the author's country, load and clean the data, and generate a world distribution map of national cooperation on the research topic of nature reserves. Select "Social Structure" on the web page finally generate "Collaboration World Map".

## 3. Results

### 3.1. Visual Presentation of the Research Topic

In the co-occurrence map, the 60 resulting topic keywords were grouped in three different clusters (Cluster 1: 28; Cluster 2: 24; Cluster 3: 8) (Table 1), and sized proportionally to their frequency of Co-occurrence (Figure 1).

**Table 1.** Clusters and relative keywords resulting from the co-occurrence analysis of keywords.

| Cluster 1 (Red) | | | Cluster 2 (Green) | | | Cluster 3 (Blue) | | |
|---|---|---|---|---|---|---|---|---|
| K | O | T | K | O | T | K | O | T |
| Stakeholder | 4537 | 48,413 | Area | 2868 | 31,909 | Data | 1256 | 14,309 |
| Study | 2671 | 29,549 | Management | 2552 | 29,447 | Model | 1070 | 12,228 |
| Approach | 2098 | 24,603 | Community | 1549 | 17,887 | Information | 985 | 11,394 |
| Process | 1860 | 21,581 | Conservation | 1330 | 15,515 | Tool | 928 | 11,313 |
| Analysis | 1647 | 18,835 | Impact | 1277 | 14,954 | Assessment | 859 | 10,216 |
| Development | 1613 | 18,920 | Change | 1126 | 13,569 | Decision | 787 | 9466 |
| Paper | 1552 | 17,531 | Resource | 1121 | 13,353 | Application | 605 | 7389 |
| System | 1396 | 16,334 | Species | 1003 | 11,279 | Risk | 556 | 6333 |
| Research | 1383 | 15,711 | Benefit | 849 | 9973 | | | |
| Framework | 1104 | 13,208 | Region | 781 | 9163 | | | |
| Policy | 1096 | 12,851 | Effect | 772 | 8961 | | | |
| Challenge | 1052 | 12,404 | Ecosystem | 769 | 9501 | | | |
| Nature | 1045 | 11,175 | Effort | 742 | 8579 | | | |
| Practice | 1010 | 11,765 | Year | 732 | 8234 | | | |
| Issue | 1001 | 11,682 | Protected Area | 717 | 8333 | | | |
| Context | 936 | 11,148 | Conflict | 644 | 7460 | | | |
| Role | 852 | 9913 | Protection | 617 | 7414 | | | |
| Interview | 770 | 8874 | Biodiversity | 610 | 7571 | | | |
| Problem | 714 | 8448 | Person | 610 | 7167 | | | |
| Project | 702 | 8203 | Perception | 583 | 6788 | | | |
| Case | 675 | 7878 | Site | 574 | 6597 | | | |
| Understanding | 674 | 8142 | Ecosystem Service | 483 | 5844 | | | |
| Way | 660 | 7695 | Marine | 426 | 5099 | | | |
| Government | 572 | 6656 | Effectiveness | 414 | 4947 | | | |
| Article | 544 | 5876 | | | | | | |
| Relationship | 535 | 6154 | | | | | | |
| Sustainability | 510 | 6109 | | | | | | |
| Concept | 485 | 5716 | | | | | | |

K: Keywords; O: Co-occurrences (frequency of keywords); T: Total link strength (the cumulative strength of the links of an item with other items).

Cluster 1 included the core keywords "Stakeholder and Nature", whose themes were discussed through the study of the relationship between nature and stakeholders, such as

"Framework", "Policy", "Challenge", and "Practice", each of these words Co-occurrences more than 1010 times. Secondly, it also involves "Analysis", "Role", "Interview", and "Understanding" from various stakeholders discuss the relationship between them, among them, the Co-occurrences of "Interview" was 770 times, and the total link strength reached 8874 times.

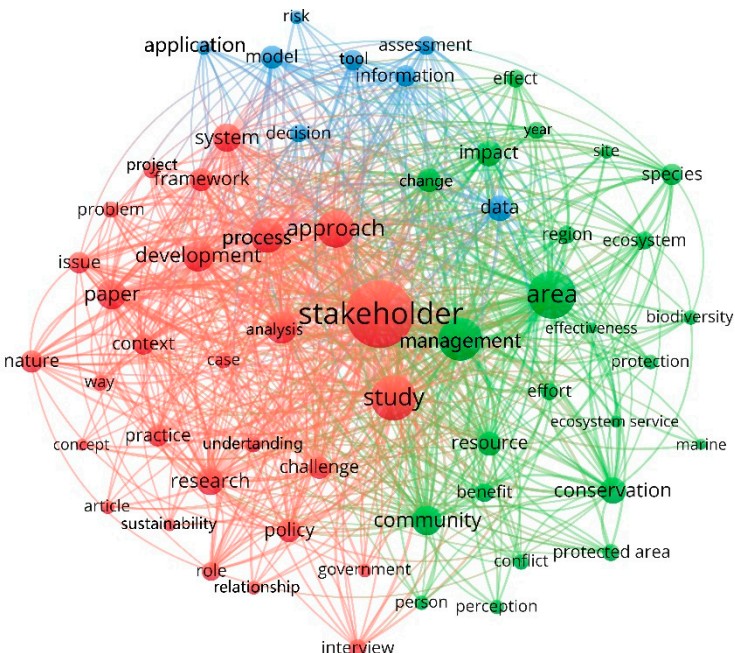

**Figure 1.** Co-occurrence network map of keywords in the global scientific literature on NPAs and stakeholders. The size of each keyword (node) in the network is directly proportional to its number of occurrences in the documents analyzed. Colors indicate clusters to which keywords are univocally assigned based on their reciprocal relatedness.

Cluster 2 includes the core keywords "Protected Area", showed keywords relating to conflicts between NPAs and communities or regions, such as "Management", "Community", and "Conflict", the Co-occurrences was 2552, 1549 and 644. Secondly, it also involves "Conservation", "Resource", "Species", "Region", "Ecosystem", "Protection", "Biodiversity", "Ecosystem service", "Marine" and other aspects of conservation and biodiversity, among them, "Conservation" has the highest Co-occurrences (1330), while "Marine" the lowest Co-occurrences (426).

Cluster 3 includes the words "Model", "Information", "Tool", "Assessment", and "Application", the way research on NPAs.

Visually (Figure 1), Cluster 1 (in red) shows a high degree of overlap with Cluster 2 (in green), because some key words were placed between the network regions of the 2 clusters, the area of the entire network map is larger. Cluster 3 (in blue) is located at the top of the entire network graph, with less overlap with Cluster 1 and Cluster 2, and less area. The intersection area between Cluster 1 and Cluster 2 is highly interconnected, showing that the distance between "stakeholder" and "management" is relatively close, reflecting the stakeholders of NPAs and focusing on research on "management".

Overall, the entire right part of the network diagram in Figure 1 shows the conflict between NPAs and stakeholders, involving a wide range of impacts, such as "species", "region", "ecosystem", "ecosystem service", "biodiversity", "marine", and "community"; while the left part shows the research on natural resources, which mostly involves the theoretical level. From a socio-ecological perspective, NPAs faces the difficult task of balancing the use of natural resources and conservation by stakeholders, a goal that is more likely to succeed through ecosystem-based collaborative management, community

engagement. Achieving the integration of human factors into nature conservation and natural resource management is a key issue for sustainable development.

Figure 2 shows an overlay visualization map based on the year the document was published, providing a temporal perspective for explaining the co-occurrence network map of keywords, and the overlay shows the temporal differences in their co-occurrence [32]. The distribution of keywords along a temporal gradient helps to understand the evolution of scientific research between NPAs and stakeholders, node colors are determined by the average time of the year in which each keyword is located, and to identify the latest topics and research paths. Overlaid visualizations show that more recent attention has been paid to human attitudes towards NPAs, such as "interview", "perception", and "ecosystem service".

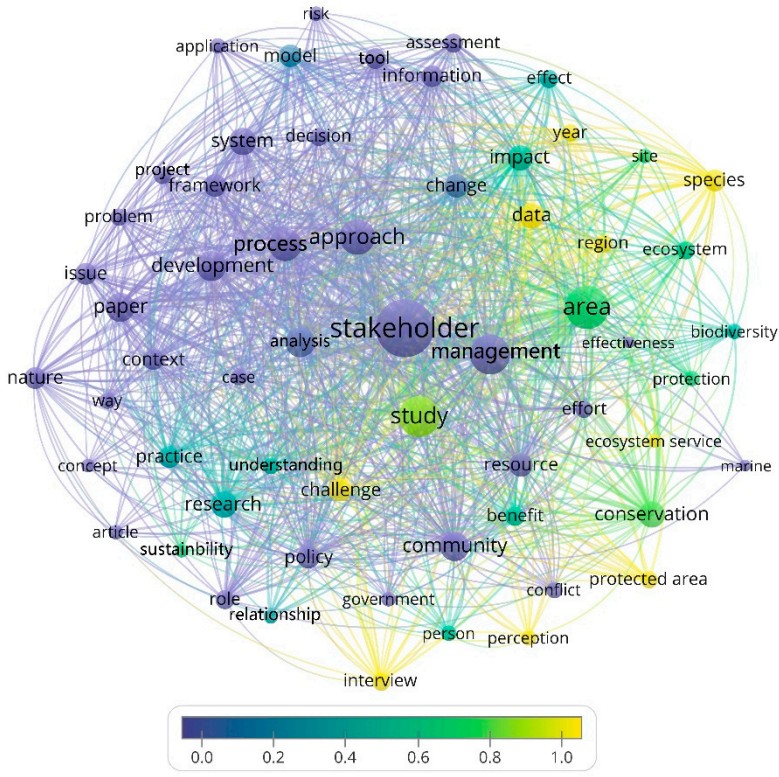

**Figure 2.** Overlay visualization of the co-occurrence network map of keywords. Keywords are represented based on the average year of publications of documents they occur in, on a color gradient from blue (older publications), to purple (publications equally distributed across the timespan 2000–2021), to yellow (more recent publications).

In addition, other keywords, such as "community", "conservation", "management", and "stakeholder", were evenly distributed over the 0–1 (Publication time 2000–2021) range frame, indicating that the rate of appearance of NPAs in the stakeholder literature is highly uniform. According to the point of view of the score index, the research on "stakeholders of NPAs" was more concentrated after 0.6 (2013 year), from the original research object of NPAs to a single natural resource, more turned to the impact of human beings on it.

### 3.2. Visualization of the Number of Published Papers and the Closeness of National Cooperation

By searching the 5584 articles in the core collection of WOS, selecting the publication year and the origin or nationality of the author of the article, Figures 3 and 4 are obtained. As can be seen from Figure 3, thematic research articles on stakeholders of NPAs are increasing year by year, especially after 2013, with an average annual publication volume of more than 335 articles. Among the scientific literature collected for 21 years, the number of studies in 2001 was the least (37) and the largest in 2021 (564). In addition, through

the bar chart of the number of published articles per year, it is found that with the fitting coefficient R$^2$ > 0.979, the prediction effect is better. Therefore, it can be predicted from the trend line that by 2025, the annual number of research papers on this topic will reach about 732, and even by the middle of this century, it will reach about 2376.

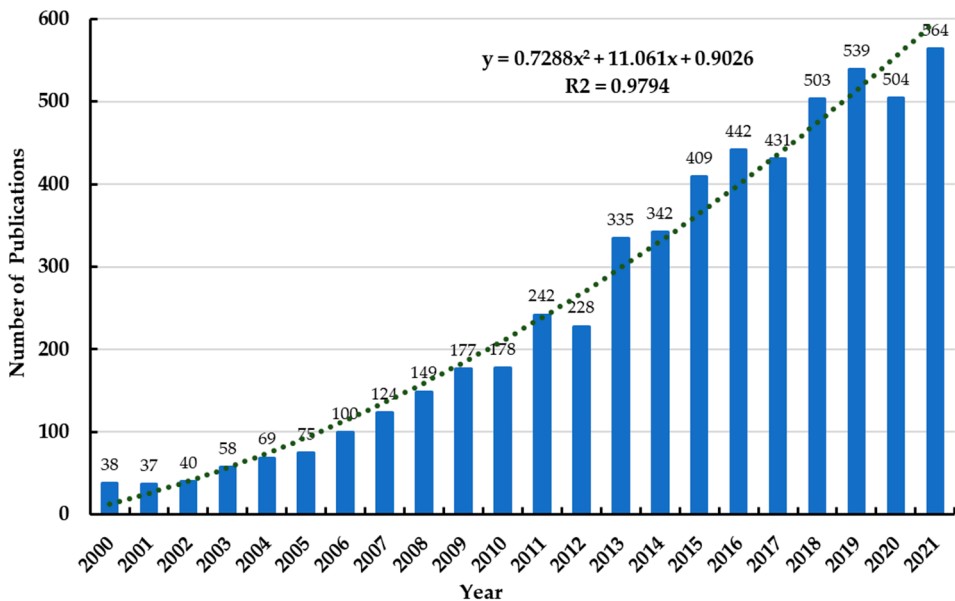

**Figure 3.** Number of Articles Published by NPAs Stakeholders from 2000 to 2021.

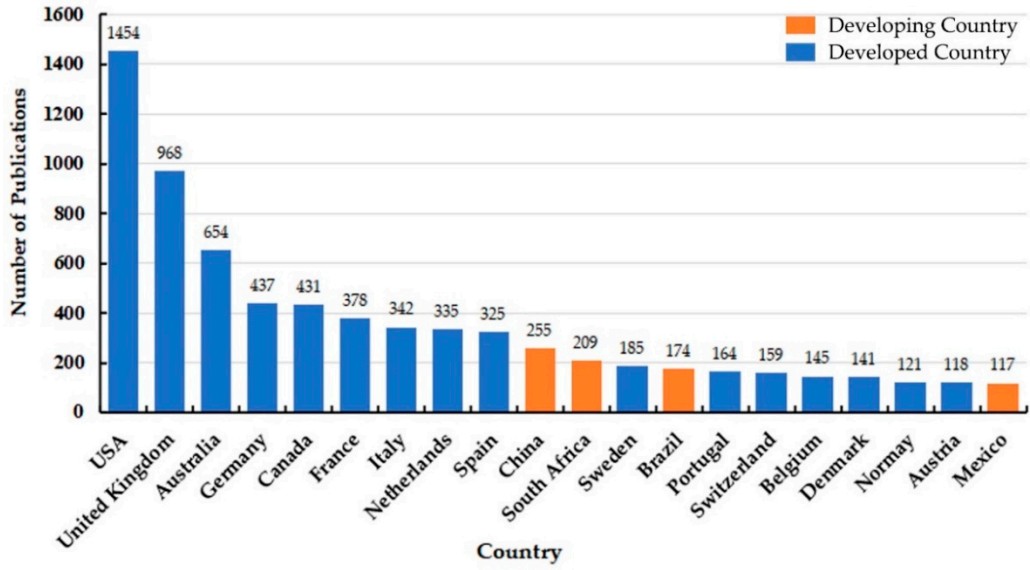

**Figure 4.** Top 20 countries with the total number of publications on NPAs from 2000 to 2021.

Figure 4 shows the top 20 countries by total publications on NPAs stakeholder research over the past 21 years. Among them, the total number of published papers in the USA is far ahead of other countries with 1454, followed by United Kingdom (968) and Australia (654), and Mexico has the lowest total number of posts (117). In terms of the degree of development of the country, the number of developed countries accounts for 4/5 of the top 20, while developing countries (China, South Africa, Brazil, and Mexico) only account for 1/5. In addition, it can be seen from the figure that the total number of developing countries (755) is lower than that of the USA or United Kingdom, only accounting for about 10.79% of the total number of published documents, and the actual situation may be lower than this value.

By loading the Bibliometrix R-package, the default system parameters are used to draw the cooperative relationship between countries (Figure 5). The color gradient represents the number of published articles: the darker the color, the more the number of published articles. It can be seen from the figure: the African and Asian regions in the figure, there are more gray regions, followed by South America and North America, which means that these countries have not published articles on the stakeholders of nature reserves or cooperated with other countries on WOS. Examples of such countries are North Korea, Myanmar, Georgia, Azerbaijan, United Arab Emirates, Yemen, Syria, in Asia; Libya, Mali, Sudan, Congo, Zambia, and Somalia in Africa; Greenland, Cuba, Haiti, and Honduras in North America; and Paraguay, Guyana, etc., in South America. It is not difficult to find that these regions have one thing in common, almost all of them belong to developing countries, with a large number of poor people, high dependence on agricultural production, and an underdeveloped market economy, which will inevitably affect the NPAs development process to a certain extent. At the same time, almost all European countries and regions regularly publish articles on NPAs stakeholders on WOS and cooperate closely with other countries.

## Country Collaboration Map

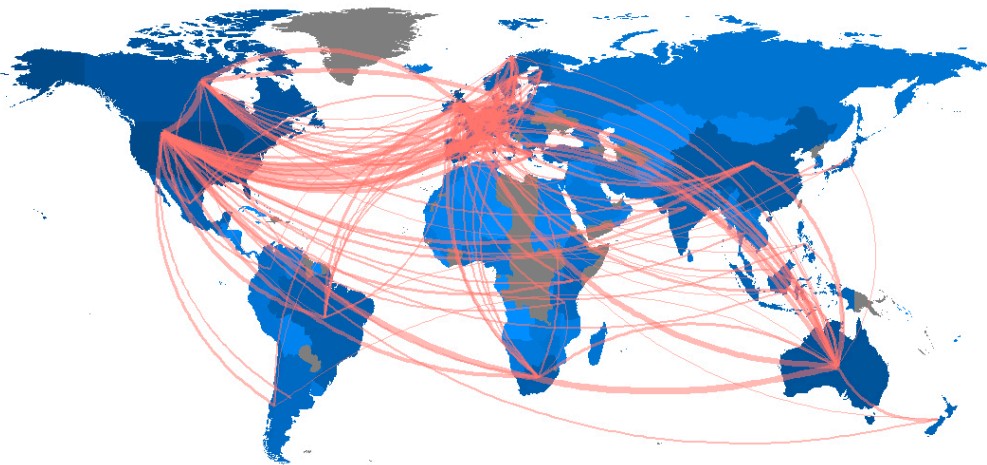

**Figure 5.** World map of country publications and collaboration on NPAs research topics. The color gradient represents the number of published articles, the darker the color, the more the number of published articles.

On the other hand, through the analysis of the closeness of cooperation between countries and countries, the cooperation between the USA and other countries is the most prominent, especially the cooperation with the United Kingdom. Secondly, Australia, Canada, Germany, and China cooperate closely with each other. It is worth noting that, as the largest developing country in the world, China has actively performed research in this field, which reflects the efforts made by the China in the field of global NPAs.

In a word, the above analysis shows that in terms of research on stakeholders of global NPAs, the number of published articles is increasing year by year, and there are more and more studies in this field; the authors of the articles are mostly from developed countries, and the research in this field is more, with less research coming from developing countries.

### 4. Discussion

Based on the findings in Figures 1–5 and a careful reading of key literature, we conduct a specific analysis of the stakeholders involved in NPAs. We then summarize the general types and composition characteristics of stakeholders, and discuss and analyze hot research issues, such as resource utilization and protection, community, management, ecosystem,

and biodiversity. Our analysis is based on classifying the types of stakeholders and sorting out the interrelationships.

### 4.1. Approximate Types of Stakeholders

According to the stakeholder theory, any group or individual that can affect or be affected by planning objectives should be considered a stakeholder [37]. Accordingly, most scholars distinguish relevant research by nature reserve, national park, nature reserve, or other types, such as research objects, and specific distinctions are also made according to the attributes of the stakeholder's interest, the relationship between the interests, and the degree of force generated. What is more, stakeholders are classified into different groups according to the stakeholder types [38–40]. As early as 1997, stakeholders were divided into the definition, prospective, and potential stakeholders according to their three attributes of legitimacy, importance, and urgency [10,41]. Grilli et al. [42], Pelyukh et al. [43], and others divided the identified stakeholders into key, primary, and secondary. Different scholars have different classification methods for the types of stakeholders, but they all emphasize the importance of stakeholders, and those with a high degree of importance generally include government departments, community residents, concessionaires, and visitors.

Most of the identification of stakeholders in NPAs is based on three methods: literature research, interview, and questionnaire [44,45]: the latter two research methods are mainly based on a case study [2,10,11,15]. The research object of this paper was the global NPAs, so the first method is adopted to carried out quantitative research on the literature over the past 21 years. Based on Carroll and Buchholtz [46], the stakeholders were divided into three categories: "Core Stakeholders", "Strategic Stakeholders", and "Environmental Stakeholders", and nine stakeholder groups (Figure 6), including all institutions, organizations, and individuals related to the pre-development and post-maintenance of NPAs.

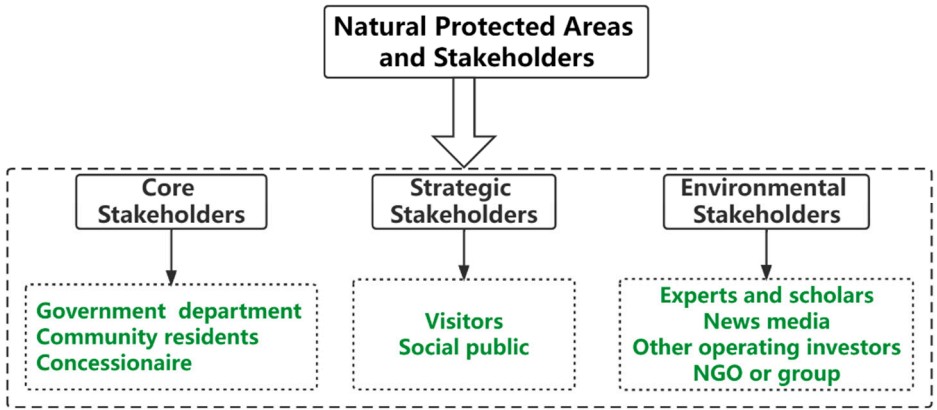

**Figure 6.** Stakeholder composition characteristics. The terms and scheme used in the present figure were according to Carroll and Buchholtz (1993) with modification.

Core stakeholders refer to groups that play a decisive role in the development, utilization, and protection of NPAs, including "Government department", "Community residents", and "Concessionaire". Among them, the Government department mainly refers to the different administrative departments in charge of NPAs. Although there are differences in different countries, they mainly focus on the "environment, fishery, agriculture, industry and commerce, tourism, taxation", and other departments. Furthermore, they also involve scenic spot management agencies and related organizations established by governments at all levels. Community residents mainly refer to ordinary residents who live in and around NPAs; and concessionaires mainly refer to enterprises that reasonably carry out business activities of related industries in NPAs in accordance with the relevant management measures formulated by government departments, including tourism developers, operators, and managers.

Strategic stakeholders refer to groups that can play an important role under certain conditions, mainly "Visitors" and "Social public". Visitors refers to individuals or groups who enter the peripheral areas of NPAs for recreational experience, including holiday tourists and sightseeing tourists, while the social public refers to the general public who participate in all other social activities in NPAs.

Environmental stakeholders refer to the social groups or institutions that have a certain influence on the creation of the land development and utilization environment and social atmosphere during the development of NPAs, including "Experts and scholars, News media, Other operating investors and Non-Governmental Organization (NGO) or group". Among them, news media refers to the publicity and reporting of relevant information on NPAs by a group or individual through paper media (Newspapers), electronic media (Radio, Television) and social media (Facebook, Tik Tok), etc. In addition, NGO or groups mainly refer to non-profit enterprises, historical and cultural research societies, tourism planning organizations, historical and environmental ecology research institutions, and academic research institutions, as well as other assistance to government departments in the management of NPAs.

### 4.2. Stakeholder Composition Characteristics

Stakeholders are guided by the goals of their own interests, and conflict games are the norm, while coordination and cooperation are the development trend. There are complex relationships among stakeholders of NPAs, involving the utilization of ecological resources, ecological protection, and ecological economic interests. By defining the rights, responsibilities, and interests of stakeholders, clarifying their mutual relationships, it is conducive to balancing the interests of stakeholders' important pathways [10].

The relationship between the stakeholders of NPAs is interactive, and their respective responsibilities and interest demands are different. Based on the stakeholder groups (Figure 6), we developed a stakeholder relationship map (Figure 7). In this map, the "government department" was responsible for the provision of employment, support, and supervision, to the "community residents" and management, protection, and development to the "concessionaire". In contrast, the "community residents" provide limited access to "visitors". The services and products of the "visitor" may also interfere with the life of the "community residents", and there is a benefit-sharing relationship between the two; the "concessionaire" provides the "visitors" with characteristic services and products, and has a relationship with the "community residents" has a coexistent relationship, but with unequal responsibilities and powers; "visitors" enjoy the good ecological environment and high-quality tourism resources in the NPAs, and at the same time obtain high-quality ecotourism experience; "other business investors" assist in the management of NPAs while operating profitably; "NGO" and "news media" supervise the status and development of NPAs, and play a role in public opinion supervision and regulation; "experts and scholars" provide theoretical reference for decision makers and participants of NPAs policies by conducting scientific research.

### 4.3. Stakeholders and Natural Resource Relationships in NPA

The contradiction between the protection and development of NPAs lies in the protection and utilization of natural resources, the core of which is the coordination of the relationship between people and natural resources [15]. With the continuous growth of the global population, the impact of human beings on the environment is also increasing, and there is an increasing demand for types of natural resources. Even if human beings have strengthened their control over the environment and improved the level of technological innovation, they are still faced with various challenges and threats, including pollution, overexploitation, encroachment, and poaching, which disrupt the state of the natural environment [47,48]. Conflicts between two or more stakeholders over resources (natural, intangible, landscape, etc.) occur frequently [45]. NPAs are rich in natural resources, including non-renewable (metallic and non-metallic minerals, fossil fuels, etc.) and

renewable resources (biological, water, and land resources), and the indigenous peoples in or around the area rely on resources for their livelihoods and for food, such as logging, picking medicines, collecting wild vegetables, honey, and firewood [49,50].

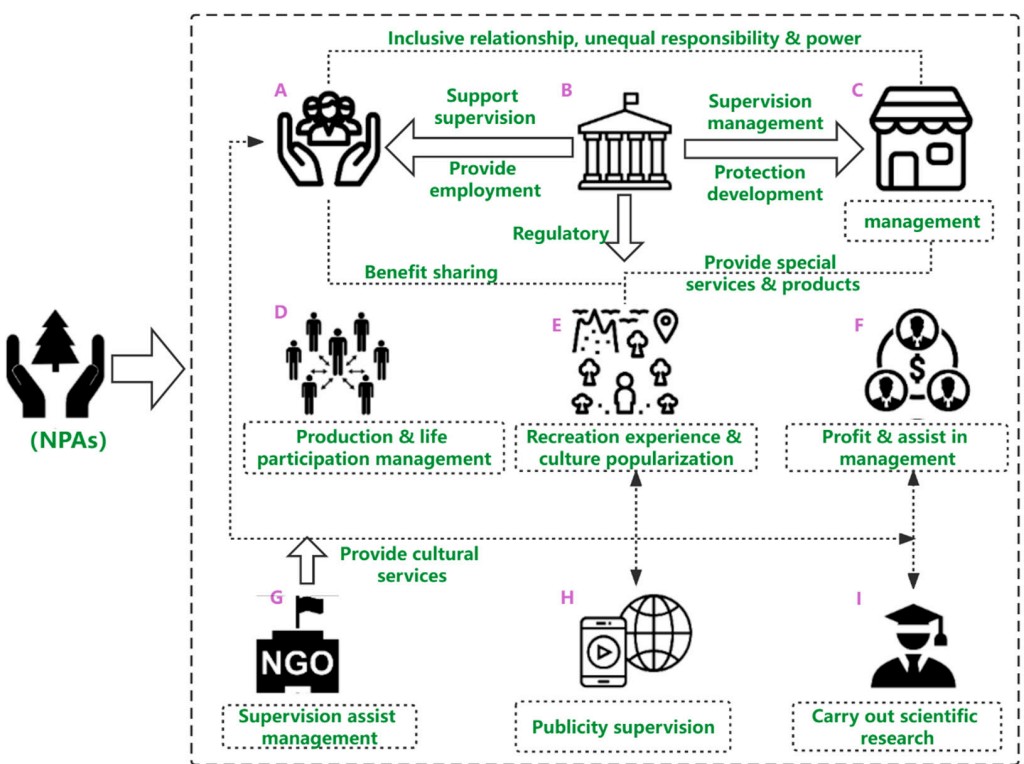

**Figure 7.** Map of the relationship between stakeholders in NPAs. (**A**)—community residents; (**B**)—government department; (**C**)—concessionaire; (**D**)—social public; (**E**)—social public; (**F**)—other operating investors; (**G**)—non-governmental organization; (**H**)—news media; (**I**)—experts and scholars.

However, from the very beginning of the establishment of NPAs, the conflict of interests of the stakeholders already exists, manifested between regions or countries. The establishment of NPAs means that in order to preserve or protect the natural ecosystems in the area, human access to land and natural resources is subject to certain restrictions or controls. For NPAs stakeholders, laws and regulations promulgated by government departments can have a substantial impact on their quality of life and livelihoods [11,51]. The economic and social problems faced by many developing countries jeopardize the effectiveness and survival of their NPAs. Rural poverty exacerbates the need to access natural resources in NPAs and increases public conflict with NPAs management [52]. The establishment of NPAs implies a reallocation of resources and rights, which tends to increase the livelihood vulnerability of local people [12,53], whose traditional livelihood strategies are often highly dependent on natural resources. In order to meet the needs of a rapidly growing population, the use of natural resources has intensified [54,55]. However, NPAs authorities restrict these activities [56], but strict regulation of natural resource use often leads to community hostility towards ecological conservation projects and conflicts with NPAs managers [12,57], this has become a key constraint on NPAs development [2].

Global demand for agricultural land is colliding with environmental protection goals, struggling to feed a rapidly growing population in the face of dwindling supplies of land, water, nutrients, and biodiversity. Food for the United Nations and agricultural organizations estimate that food production will need to increase by 70% by 2050 to feed an estimated population of 9.1 billion [58]. Faced with abundant land and forest resources in and around NPAs, conflicts often arise among stakeholders, who expect to obtain more land or forest resources for their livelihoods. Especially when the NPAs are located in poor countries, remote locations, and the economic development of surrounding

villages is lagging behind, the local industries are dominated by primary industries such as agriculture, fishing, forestry, and animal husbandry. The ability of community residents to seek alternative livelihoods is also very limited. Their production and life require certain basic resources, which may have adverse effects on the ecological environment in the region (Figure 8).

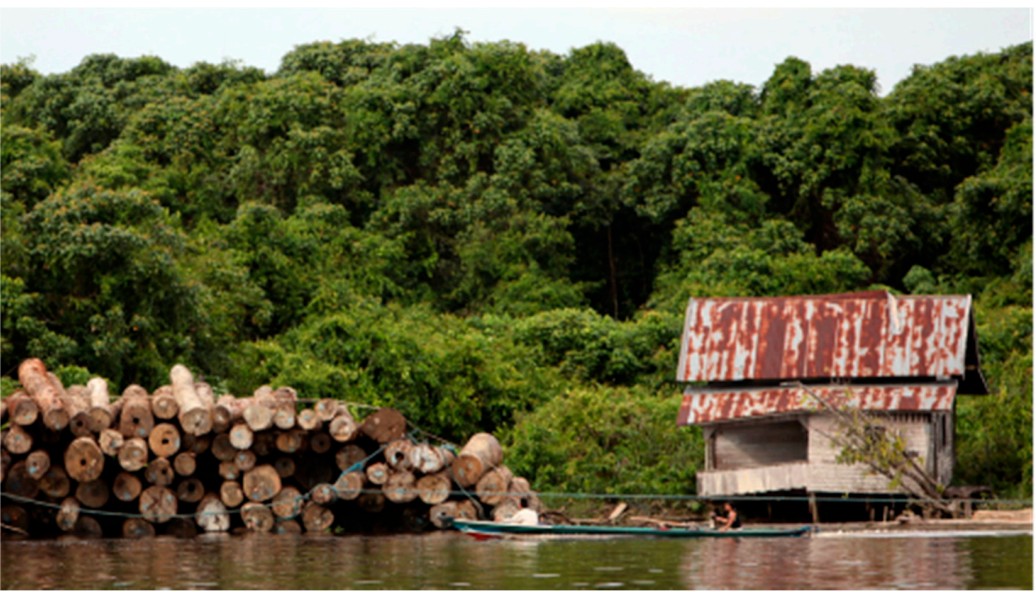

(**a**)

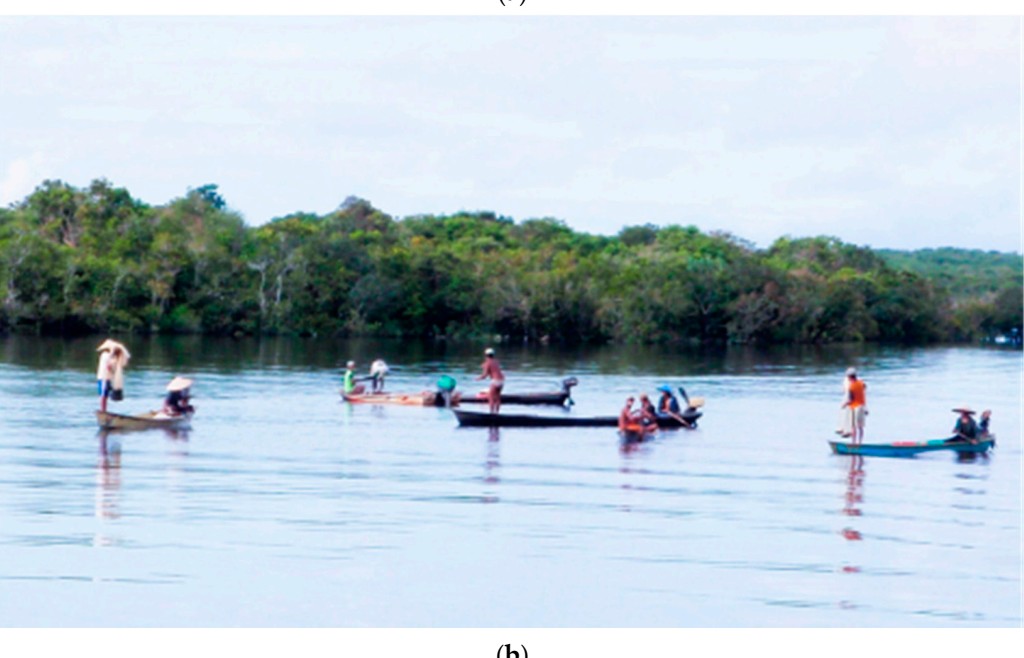

(**b**)

**Figure 8.** Community residents demand for some natural resources. (**a**) "Kalimantan deforestation and degradation 3" by DFAT photo library is marked with CC BY 2.0; (**b**) "Communal fishing" by CIFOR is marked with CC BY-NC-ND 2.0.

Therefore, the prerequisite for the long-term sustainability of NPAs is public participation and support for the protection of natural resources, timely resolution of conflicts between various stakeholders due to resources and related to the development process of NPAs and people's livelihoods.

### 4.4. The Management Relationship between NPAs and Community Residents

Worldwide, about 50% to 70% of NPAs have indigenous peoples, and most NPAs are established with the expulsion of local communities or without adequate consideration for their livelihoods. Strict protection measures often displace people and deprive them of their livelihoods, affect local economic development, exacerbate their poverty, and often lead to community conflict [14,59,60]. Socio-economic and cultural factors largely influence conservation decisions in most developing countries [61], the traditional "top-down" approach to NPAs management without the cooperation and support of community residents, especially in developing countries, has limited development prospects [62,63]. In order to alleviate the conflict between NPAs and the community and improve the participation of community residents, there is a need to establish a community co-management mechanism in the management of NPAs [49].

Community-based co-management (CBCM) refers to a negotiation process involving two or more stakeholders to determine and ensure equitable sharing of decision-making arrangements, planning, rights, obligations, and management functions in natural resource management. In addition, it is a way to involve community members in the decision making, implementation, and evaluation of conservation programs [64].

Solving resource and environmental problems through a collaborative process promotes ecologically sustainable livelihoods, most importantly, avoids the drawbacks of traditional management and governance of NPAs, and enables local residents and the state to participate in the decision-making process on an equal footing [65]. This is one of the important measures to alleviate the contradiction between humans and land in NPAs. The Fifth World Parks Congress in 2003 pointed out that community residents, mainly the poor, bear the main cost of conservation, but community residents must participate more effectively in NPAs, and especially their rights must be fully respected [66]. However, there are various prerequisites for achieving co-management, such as effective communication of clear common goals, building of trust, conflict management, and long-term government support [67]. All these studies focus on stakeholders. Izurieta et al. [68] believed that 27 indicators such as social, cultural, and economic outcomes of NPAs should be used to evaluate the results of joint management, especially the rights and interests of stakeholders.

It is undeniable that government departments play a major role in the management of NPAs, entrusting some management responsibilities to local communities and relevant stakeholders [69], striving to promote a sustainable development model, and expecting to gain the support of other stakeholders, and by cooperating to realize the hierarchical management of public participation. In order to avoid the marginalization of public interests, stakeholders can participate in decision making, management, and sharing, and conduct research together with decision makers to improve the management efficiency of NPAs. At the same time, the participation of stakeholders can be promoted to realize ecological protection and socio-economic development, maintain biodiversity conservation, and improve human well-being [70,71].

### 4.5. Development of NPAs and Synergy between Stakeholders

The protection and development of NPAs are two of the most important topics [72], and hot issues in the research field, and are important factors related to human well-being. The combination of these keywords indicates (Table 1) that stakeholders are mostly associated with communities, biodiversity, ecosystems, and marine areas, and also reflects the main contradictions in the field of study. Due to the differences in the cognition of the benefits of NPAs by various stakeholders, it will be difficult to form a synergy in the protection and management of NPAs, and it will be difficult to achieve synergy between the development of NPAs and stakeholders. This will change if stakeholders agree to, create a positive feedback effect, reduce the tendency towards self-interest, and increase the sense of altruism between the ecological and economic benefits of NPA. So, where do improvements need to be made?

The first step is to improve the management of NPA. From the visualization in Figures 1 and 2, it is easy to see that "Management" stands out among the 60 keywords, meaning that it appears more frequently in the articles, which indicates that management is one of the essential factors for the development of NPAs in the context of the study of NPAs stakeholders. However, what is the object of management? Nature or human beings? From the perspective of the development and changes of research topics over time (Figure 2), the keywords of "Perception, Person, and Interview" give the answer. In the published articles in recent years, it is not difficult to find that the research field has gradually shifted from the management of natural resources to the management of social and ecological systems. In fact, the management of the latter is the real challenge facing NPAs [11].

The second step is to increase the enthusiasm of all stakeholders to participate in the protection of NPAs. Stakeholder engagement is often cited as a fundamental prerequisite for sustainable management of natural resources, especially NPAs [11]. Stakeholders should be involved in the management of NPAs as early as possible, as such participation is essential for the development of NPAs and the formulation of conservation policies. Local communities and other stakeholders are more likely to adhere to and commit to long-term conservation strategies if the knowledge and input of local communities and other stakeholders are incorporated into NPAs decision making prior to the development and implementation of conservation policies [73]. However, the reality is that some stakeholders choose the "Ostrich mentality", they do not have a strong desire to participate. Without all stakeholder groups participating in decision making, transaction costs will increase and make it difficult to form a unified result [74]. When the specific implementation of the policy threatens the interests of a group, their needs are not met, so there may be conflicts with the managers of NPAs [12], and in the long run, a vicious cycle will be formed. Therefore, by simulating "Participatory Scenarios", it is possible to explore the choices of stakeholders for resource utilization in NPAs, and on the other hand, it is also a way to provide reference for decision makers. The participatory scenario approach is a way for stakeholders to achieve self-organization and self-actualization [15]. Moreover, it can be applied through collaboration between stakeholders to better understand the causes of changes in drivers and improve future forecasting [75]. However, it cannot be ignored that failed to involve stakeholders in the decision-making process in an equitable manner will lead to suboptimal and sometimes unethical outcomes [58].

The third step is to resolve the conflict between protecting resources and the well-being of community residents. The NPAs and its surroundings are rich in resources, including water, land, animals, rare and relict plants and endemic vegetation types, topography, and landscape, which have great social, economic, and ecological value. It is of great significance to find a way to balance the contradiction between local economic development and ecological protection [49]. In Figure 1, the core keyword "Community" of Cluster 2, revolves around "Government, Interview, Conflict, Person, Perception, and Benefit", indicating that through community-based and community residents' participation, and in the context of government, the development of NPAs protection, plays a very important role (Figures 6 and 7). In the protection and management work, the role of the community is very important. On the one hand, community residents are essential contributors to environmental protection. On the other hand, they are also users of NPAs resources, excessive use of resources will bring certain ecological pressure and danger. Since the 1980s, tensions have persisted between the need for conservation and the well-being of local communities [44,76], including in the resettlement of indigenous peoples, limited access to natural resources, lack of cultural and social institutions, and loss of livelihoods [53]. In particular, ignoring the interests of local people and excluding them from protected area planning, management, and decision making are major sources of conflict between local people and NPAs [23,59]. In addition, the environmentalists have achieved a certain degree of sustainable development of NPAs and made certain contributions to biodiversity conservation through actions such as preventing deforestation of forest resources. At the same time, this behavior has brought certain social and economic benefits to local

stakeholders [2]. Therefore, some scholars suggest that the annual income of local farmers can be increased by introducing alternative cash sources for traditional planting [49]. The livelihood problems of community residents can be solved, and the tourism industry can be developed to ensure sustainable economic benefits for local communities by cooperation of various stakeholders [77,78].

### 4.6. Analysis among NPAs Stakeholders

By sorting out the characteristics and relationships of various stakeholders, the team members in this study are based on the arrangement of a large number of documents, similar to the method of "Focus Groups", to roughly classify and analyze the characteristics of the stakeholders involved (Figures 6 and 7), without reference to a specific regional case. In general, qualitative and quantitative methods are used for inter-stakeholder analysis, including: Focus groups, Snowball sampling, Q methodology, Semi-structured interviews, Social network analysis, Knowledge graphs, and other methods for sampling and analysis [79]. Although different stakeholder case studies have produced different stakeholder compositions, but the common denominator is around core stakeholders (Community residents, Concessionaire, and Social public) (Figure 6) to carry out mainly research, supplemented by other stakeholders. Among them, the role of the government is very important, relying on national agencies, forest rangers, the army, and the police to protect NPAs from the aspects of policy, implementation, and coordination with other stakeholders [80].

Disputes among various stakeholders include the division of the use space of NPAs, the demand for natural resources, the needs of reasonable laws and regulations, and concerns about the ecological carrying capacity of NPAs. For example, the government department, experts and scholars, and NGO hope to increase the number and area of NPAs to solve the problems of protection management division, fragmentation, and isolation of NPAs, and further realize the overall protection of natural ecosystems. However, community residents believe that the establishment and protection of NPAs reduce the resources of local communities and limit their development opportunities [81], causing serious land-use conflicts [82]. Concessionaires and other operating investors expect to expand the scope of their business operations, so as to increase the fun and experience of visitors and make more profits. According to a limited stay in NPAs, visitors and the social public hope to enjoy the good ecological environment and high-quality tourism resources of the reserve and various types of ecosystem services within a certain period of time, and obtain a high-quality eco-tourism experience. During the construction and planning of NPAs, the influx of tourists from tourist areas has brought great pressure on the biodiversity and natural resources in the NPAs [45]. Government departments, experts and scholars, news media personnel, and NGO expect other stakeholders to increase their awareness of environmental protection. If the carrying capacity exceeds the ecological stress threshold, it will put greater pressure on the surrounding environment, thus damaging the original character of the NPAs. If irreversible, the degradation of NPAs will become a concern or have an unacceptable impact on natural resources and the quality of the visitor experience [83]. Government departments are well aware that through recreational and recreational activities in NPAs, the benefits of community residents can be increased, and local social and economic benefits can be improved [84]. However, this could lead to a dramatic increase in sensitive areas of high ecological value. Therefore, there are certain contradictions, supervision, and cooperation among the various stakeholders of NPAs. Solving the existing contradictions is conducive to jointly safeguarding the sustainable development of NPAs.

### 4.7. Source of the Article behind the NPA Stakeholder Study

While it has been established that NPAs conflicts have a strong geographic component, previous research on NPAs conflicts had often focused on assessing the underlying causes of NPAs conflicts without considering how these conflicts have changed in the context of development [69]. Therefore, we guessed whether the research background in this field is

related to the degree of development of the country through the number of sources (Country) and global distribution characteristics (Figures 4 and 5) published in the literature, and the results are consistent with expectations. Most of the research articles are concentrated in the world's top-ranked developed countries. For example, the United States is far ahead in terms of the total number of publications, followed by United Kingdom and Australia. While China, a developing country, has an active research program comparable to many developed countries in this field. The experience of cooperation reflects China's contribution to the research of global NPAs.

Because the distribution of the vast majority of NPAs highly overlaps the geographic distribution of poor areas, NPAs face a strong conflict between conservation and development [21,85]. Soliku and Schraml [69] argued that the types of NPAs conflicts, why they occur, where they occur, and how they are managed vary across developed and developing countries and are determined by geographic location and specific socioeconomic and cultural contexts. The relationship between NPAs and poverty is a long-debated issue in academic and policy circles [86]. In developing countries, where high levels of poverty and social inequality [87] are commonplace and where most of the world's biota [88] and ethnocultural diversity are found, research on NPAs is lacking. The developed countries have perfect policy theory and advanced scientific research technology, which has a great advantage. Finding new ways to reconcile socio-economic well-being and natural sustainability is critical for contemporary societies, especially in tropical developing countries [2].

## 5. Conclusions

We conducted a quantitative review of the global scientific literature on NPAs stakeholders over the past 21 years and explored the research hotspots, research trends, publication time, the number of publications, and countries of publication of articles in this field. Our results show that the top three keywords used by NPAs studies are "management", "community", and "conservation". However, there is a huge unbalance in the number of studies originated from the developed countries and those from the developing countries. Of the top 20 countries with the most published studies, 16 were developed countries. Furthermore, studies from these developed countries accounted for 89.21% of all the publications. In addition, over the 21-year study period, the focus of NPAs studies showed a gradual change, shifting from environmental diversification to human social relations.

Our study is based on the bibliometric analysis of the NPAs literature. The method is based on published results and their citations in the WOS core database. The Bibliometrix package is loaded in R. We selected the WOS core database in this study as only one database could be selected for analysis when encountering multiple databases such as WOS core database, Scopus, Dimensions, and PubMed. However, some countries or regions have their own article databases, and the research results are in their own languages. Due to the limitations of the WOS core database, this method only displays articles published in English. Hence, our analysis on the NPAs literature should be viewed as studies published in English only.

Faced with the unbalanced development of NPAs between different countries or regions, it is necessary to implement interdisciplinary, cross-country, and cross-regional cooperation, and coordinate the development of NPAs with stakeholders. The concerted efforts of all stakeholders are essential to achieve the global sustainable development goals for NPAs.

**Author Contributions:** Conceptualization, Y.Z. and P.W.; methodology, J.X. and Y.Y.; writing—original draft preparation, Y.Z.; writing—review and editing, Z.Y.; visualization, Y.Z. and M.T.; funding acquisition, P.W. All authors were committed to improving this paper and are responsible for the viewpoints mentioned in this work. All authors have read and agreed to the published version of the manuscript.

**Funding:** This research was funded by the Key R&D Program of Hubei Province, grant number 2020BCA081.

**Institutional Review Board Statement:** Not applicable.

**Informed Consent Statement:** Not applicable.

**Data Availability Statement:** Data available on request due to restrictions.

**Conflicts of Interest:** The authors declare no conflict of interest.

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
