# Peer review of "What Is the Relationship between Natural Protected Areas and Stakeholders? Based on Literature Analysis from 2000–2021"

_forests, doi:10.3390/f13050734_

Round 1

Reviewer 1 Report

This paper provides an interesting bibliometric analysis on the stakeholders analysis from NPA. However, there are some issues which need to be addressed in order to make it suitable for publication.

On line 261 the authors mention England while Figure 4 specifies United Kingdom.

On lines 269-270 there is a sentence without sense (Limitations....).

The discussion section is written with single space paragraphs.

On line 516 is is Snowball gampling or sampling?

On line 579 there is an extra "f" letter which does not belong there

The conclusions are rather short and weak, they need to include information about the theoretical and practical implication of this study, its main limitations as well as future research directions. The final sentence also beed to be rephrased.

Reviewer 2 Report

This is a beautifully written manuscript presenting a quantitative review of the global scientific literature on Natural Protected Areas (NPAs) stakeholders through a well used bibliometric approach.

Apart from several typos that have to be corrected and the need for a final English proofreading, I have no other comment to provide. This is well carried out study that may be published as it is. 

Just a comment about further research. Would be really interesting to distinguish between terrestrial NPAs and Marine protected areas (MPAs). This would provide useful insights for informing Biodiversity Strategies.

Reviewer 3 Report

Such systematics studies are very important and hard to be conducted. It is caused by the high sensitivity of the correctness of the results to the methodological aspects of the research. So, if the sampling design is wrong, the whole manuscript will need to be re-written or even re-thought. Therefore, I have first looked at the Material and Methods.

To my regret, the sampling design has been wrongly planned. Therefore, many potential records in databases were obviously missed. I would not like to reject completely this submission because I see that after re-performing the sampling, this study would be much better than now. My comments on the methods are below.
1. The first large mistake is avoiding databases SCIE and ESCI, where results of such studies can be published. Not only SSCI and CPCI-SSH contain these data.
2. Then, if I understand correctly (but I am sure that it is so), this research is of global scale, not focused on Chinese Protected Areas, since Fig.4 presents data about various countries around the world. Therefore, there is a significant mistake to use ONLY the following key words "Natural Protected Areas", "National Park", "Natural Park", "Nature Reserve" in the search of bibliographic records. Please, note that the search tools act simply, especially if you use quotations. For instance, if you use "Natural Protected Areas", the search will not include studies, where the sentences "Natural Protected Area" [no "s" at the end], "Protected Areas" [no the word of "Natural"], etc. exist. Moreover, if we say about protected areas, there are other categories of PAs in various countries, like "sanctuary", "natural monument", "wildlife reserve", "conservation area", etc. Many studies have been missed because you didn't take into account these PA categories. Therefore, my suggestion is to use the following key words: "protected area*" [asterisk allows to find both "... area" and "... areas"], and then all possible types of protected areas with an asterisk at the end, i.e. "natural park*", "national park*", "sanctuar*", "reserve*", etc. together with key words related to the stakeholders (exactly this word not always used in the title, key words or abstract - you may find its synonyms).

After re-building the content amount, I would be glad to review this manuscript again.

Minor comment: In Fig.4, are you sure that you don't confuse colors for developed and developing countries?

Round 2

Reviewer 3 Report

Dear Authors,
Thank you very much for the improvement (revision) of the research design in terms of the search tools used to find all records meeting the established criteria. Now, there are much less "white gaps" can be found in the search.
Concerning the responses to my previous comments, I would like say only about the Protected Area categories. Yes, without doubts, the source https://www.protectedplanet.net/ is of high importance and of global scale. However, there are many omissions in this data set. For example, when personally I worked with this database, many Russian, Ukrainian, Kazakhstani (and other countries of the former USSR), Chinese, South American (in general for many countries) Protected Areas are absent in this database. In addition, no many authors use IUCN classification of Protected Areas, by preferring national titles of Protected Area categories, where (for instance) many natural monuments can be classified to V, IV, or III categories of IUCN classification of Protected Areas. This specific feature needs to be taken into account, when we work with such materials. 
But now  let me say about the manuscript. It became better now. However, I would like to provide some suggestions.

In the Introduction, some aspects need to be re-written or clarified. For instance:
In line 35, the reference(s) is(are) needed to support the sentence ended by the following phrase "... has been added in the last decade".
Line 49: change "whose" on "which"
Line 51: delete "is"
Lines 51-52: By saying about NPAs, we should not forgot also about poaching (e.g. https://dx.doi.org/10.24189/ncr.2020.006), illegal trades of the production. It should be also mentioned. In addition, as it is known, in publications, stakeholders can be used as research objects, but they cannot be named as exactly "stakeholders". Often, these are resident people living around or even within NPAs (e.g. here https://dx.doi.org/10.24189/ncr.2021.040, resident people (who can be considered as stakeholders) living around the Khangchendzonga Biosphere Reserve provide a knowledge about threatened birds protected in this NPA).
The sentence in lines 56-57 is slightly connected with my previous comment. So, this is not always correct that NPAs are situated in "underdeveloped areas" (line 57). For instance, many studies show that even within Protected Areas, cases of poaching and illegal trade affect NPA and biodiversity within them. For instance, the illegal trade of sockeye salmon (fish, as the main food for brown bears) affect population parameters and behavior of brown bears in a Protected Area of national level importance (see https://dx.doi.org/10.24189/ncr.2021.025). The next sentence is also needed to be supported by reference(s) (as well as other strong statements in the Introduction). It is indeed the problem when the ownership issues appear. For instance, if there are conflicts between wildlife managers and resident people (e.g. https://dx.doi.org/10.2305/IUCN.CH.2013.PARKS-19-1.JSA.en, https://dx.doi.org/10.3986/AGS.895), as well as between researchers and a tourism (e.g. https://dx.doi.org/10.18306/dlkxjz.2020.12.012).
Therefore, I warmly ask you to a bit re-write these sentences by taking into account these specific features related to NPAs. In addition, I think that in the Introduction, each strong statement should be supported by the recently published references (you may use some ones cited above). This will demonstrate the relevance of the present systematic review on international scale.
Line 71: Please, note that the aim can be only one (not twofold), while many research tasks can be established to each the formulated aim. Please, re-write this sentence by stating a certain aim and establishing research tasks. Then, the indication of methods (what did you use during the research, like software (line 71), during which period (lines 77)) should be included only to the Materials and Methods. Here, please, indicate what did you want to do to reach the aim. All details about the methodology should be moved to Materials and Methods.

Line 94: I hope that the indication of “Stakeholders*” is the mistake in the text because only “Stakeholder*” (without "s" as a plural designation) can be used. Otherwise, you missed many studies, where the word “Stakeholder” was used without the word of “Stakeholders”. As we know, the asterisk at the end designates that the search will involve all records, where this piece (rightward from the asterisk) of the word will be found, i.e. if we use "find*", we will obtained results where such words will be found as "finder", "finding", "find", etc. 
Line 97: This is meaningless to use the key word Protected Areas With Sustainable use of Natural Resources*, if you don't use quotations (like it is shown in the text of the manuscript). Without quotations, you will obtain records, where each word from the sentence (Protected Areas With Sustainable use of Natural Resources*) occurs. In addition, if you have already used the key word Protected Area*, there is no need to use the key word of Protected Areas With Sustainable use of Natural Resources*, because this will be already included in records.
Line 101: I guess, you have also extracted "Date" (if I remember correctly, or it is "Year") from each publication because in Fig. 3, you use years at the axis X.
Line 107: VOSviewer should be cited as reference. If it is not possible, please, add at least the URL-link of this source.
Line 124: I think that you have used R software, but not R language. In addition, please, add the reference supporting this source. Also, each R package should also be supported by the reference.
Line 134: Figure 5 should not be cited before Fig. 4, Fig.3, Fig.2, and others! In addition, each figure should be placed into the text immediately after its citing in the manuscript.  Therefore, place Fig. 5 after line 134 by re-naming it as Fig.1, or delete this citation in line 134.

Lines 137-140: These sentences represent the methodology by designating the algorithm used. Actually, this is the information for Materials and Methods. Please, move the text to the relevant section.
Lines 153-155 represent rather the shortened discussion of the obtained data. I suggest to move this paragraph (with its extension) to the section Discussion.
Line 162: "a high degree of overlap" - please, indicate what degree (in percents?) is meant?
The sentence in lines 166-169 is the part of Discussion. It should be moved there. Here, the most important for the section Results is an indication of the frequency (degree) level of relationships between key words, why the question WHY? (it was found) should be answered and explained in the section Discussion.
Lines 182-184 should be included to the figure caption.
Figure 3 (line 200): this is a bit unclear how the date can be 2014.6 or 2015.4. They should be 2014, 2015, etc. In addition, I guess that Fig. 3 shows only the range between 2014 and 2016. However, the authors used the much larger period, from 2001 to 2021. It should be reflected in Fig.3! Otherwise, please, delete this illustration and replace it by another visualization of these results.
I don't understand... I see two Figures 3 (it should be corrected). Well, in the second Fig.3 (line 215), if I understand correctly, the cumulative sum is shown. However, it is very important to demonstrate the number of publications per year! Please, add two graphics on the same plot. First of them will demonstrate the cumulative sum of records (like it is shown now), while in the second one (maybe, being represented by column graph), please, show the number of records per year. Otherwise, the text in lines 209-210 is speculative, because we don't see these results in (second) Fig. 3 (line 215).
In Fig.5 (line 254), it is not understandable, what does mean "Legend". What parameter is shown? Please, replace the word Legend to designation what does it mean. Otherwise, please, explain it in the caption of the figure.
The whole subsection "Approximate Types of Stakeholders" is not the section Results. This is partially Material and Methods (terminology), and partially Discussion. This sections should be moved completely or by parts in other sections.
Line 282: Am I right that Fig. 9 is based on publication of  Carroll & Buchholtz (1993)? If so, it is not your Results, too. Therefore, besides the moving the text to other sections, please, in the caption of Fig. 9, add statement that this scheme used according to  Carroll & Buchholtz (1993) or according to Carroll & Buchholtz (1993) with modifications. This is methods, if I understand correctly.
Lines 323-337: if you say about certain relationships between... key words (?), please, refer to Figure where these relationships are present. In addition, the level of relationship should be mentioned each time, when you say about that. Otherwise, it sounds like speculative statements because they are not supported by any information.
The subsection "Stakeholders and Natural Resource Relationships in NPA" looks to be Discussion, but not results. Here, you discuss the obtained results in light of the literature. This partially refers to the previous sub-section (3.4). In general, please, move all fragments with the discussion in the corresponding section (Discussion).
The same suggestion concerns the next sub-section "The management relationship between NPA and community residents".

The present version of the Discussion looks relatively well. But now I don't sure how it will be looked after all modifications recommended by me. If possible, I would like to say about the updated version of Discussion (and other sections, too), but not now. But now I see that the authors have appropriate knowledge of suitable and relevant literature.

Finally, the section Conclusions should be a bit re-written. Please, note that no repetition with other sections should be here. For instance, the sentence "We  found  management is  the  keyword  with  the  most  occurrence, followed  by “community”, “conservation”, “ecosystem”and “biodiversity”" (lines 581-582). There is no need to say this, because I (as a reader) can find it in the section of Results. Please, in Conclusion, say what does it mean in terms of relationships between NPA and stakeholders. What recommendations can be made based on these results and their discussion? It would be much better.

I hope that the authors will not be discouraged by a large amount of my comments. I see that the study is interesting and important. But now it needs to be improved.

Author Response

请参阅附件
